# Shape- and Size-Controlled Palladium Nanocrystals and Their Electrocatalytic Properties in the Oxidation of Ethanol

**DOI:** 10.3390/ma14112970

**Published:** 2021-05-31

**Authors:** Seokhee Lee, Hyeongkyu Cho, Hyeon Jeong Kim, Jong Wook Hong, Young Wook Lee

**Affiliations:** 1Energy & Environment Division, Korea Institute of Ceramic Engineering and Technology (KICET), Jinju 52851, Korea; lsh@kicte.re.kr (S.L.); hkcho@kicet.re.kr (H.C.); 2Department of Chemistry and Research Institute of Natural Sciences, Gyeongsang National University, Jinju 52828, Korea; jane9598@naver.com; 3Department of Chemistry, University of Ulsan, Ulsan 44610, Korea; 4Department of Education Chemistry and Research Institute of Natural Sciences, Gyeongsang National University, Jinju 52828, Korea

**Keywords:** ethanol oxidation reaction, Pd nanoparticles, size control, electrocatalytic activity

## Abstract

Pd nanoparticles (PdNPs) were synthesized in an aqueous environment via the reduction of K_2_PdCl_4_ by a surfactant under a high temperature. Highly monodisperse spherical PdNPs and multi-pod PdNPs with a controlled size ranging from 18 to 50 nm were prepared in high yields by varying the concentration of cetyltrimethylammonium chloride. The structural and optical properties of the synthesized Pd NPs were characterized by transmission electron microscopy, X-ray diffraction and UV–vis spectroscopy. The spherical and multi-pod PdNPs exhibited catalytic properties that were unique to their size and shape and presented efficient electrocatalytic activities toward the ethanol oxidation reaction.

## 1. Introduction

Palladium nanoparticles (PdNPs) have recently attracted growing attention due to their effective catalytic abilities in a range of reactions (e.g., the Suzuki coupling reaction), in addition to their electrocatalytic activities in fuel cells, and their potential for application in hydrogen storage materials [1,2,3,4,5,6,7]. In particular, PdNPs are widely used as an electrocatalyst for anode reactions of fuel cell systems, which are clean and sustainable energy converting system, generating electricity form chemical fuels. However, several factors such as high cost, poor stability hinder the practical application of fuel cell systems. Recently, PdNPs with controlled shape and size showed enhanced electrocatalytic performances for electrochemical fuel oxidations, which signifies the importance of preparation of PdNPs with desirable shapes and sizes. In terms of the preparation PdNPs, control of the NP shape and size is essential to allowing their application in a range of fields [8,9,10,11,12]. Indeed, a number of routes for the controlled syntheses of PdNPs have been reported, including the polyol process, an aqueous phase synthetic route and template-assisted synthesis. Control of NPs with the size and shape also appears to be important in terms of controlling their crystal growth and reduction rate. This can be performed by varying the concentrations as the metal precursors, the use of capping agents and stabilizers, the application of reducing agents and variation in the solvent, temperature, pH and external ionicity [13]. Such size and shape control is also important in the context of catalytic reactions. Among the various reported shapes, the multi-pod structure has been shown to exhibit an effective high surface area, which renders it applicable in the catalytic alcohol oxidation reaction [14].

In this study, since the shape and size of palladium nanoparticles can be controlled by changing only the concentration of surfactant and K_2_PdCl_4_, the proposed synthesis method in this work is more facile compared with previously reported methods for controlling the shape and size of palladium nanoparticles. Thus, we herein report the preparation of spherical PdNPs (PdSNPs) and size-controlled multi-pod Pd nanoparticles (PdMNPs) with a high surface area through variation in the quantity of added cetyltrimethylammonium chloride (CTAC). The electrocatalytic properties of the prepared PdNPs and PdMNPs are then examined in terms of the ethanol oxidation reaction. The catalytic performance, shape, crystal structure and size of the PdNPs are confirmed by ultraviolet–visible (UV-vis) spectroscopy, infrared (IR) spectroscopy, transmission electron microscopy (TEM), scanning electron microscopy (SEM), cyclic voltammetry (CV), chronoamperometry (CA) and X-ray diffraction (XRD).

## 2. Experimental

K_2_PdCl_4_ and cetyltrimethylammonium chloride (CTAC) were acquired from Sigma Aldrich, St. Louis, MO, USA. An aqueous solution was used as a solvent for all reagents, and Milli-Q water with a resistance of 18.3 MΩ·cm was used to prepare the aqueous solution.

To synthesize PdSNPs and PdMNPs, a 5 mM aqueous solution (1 mL) of K_2_PdCl_4_ was added to the surfactant CTAC (5 mL of 10, 30, 50 or 80 mM CTAC) and 1 mL of 5 mM K_2_PdCl_4_ aqueous solution (1 mL). The solution added to the vial was sealed and placed in a 90 °C drying oven for 1 h. PdNP was washed several times with ethanol (Sigma Aldrich, St. Louis, MO, USA) and distilled water using a centrifuge (10,000 rpm for 5 min) (Thermo Fisher Scientific Inc., Waltham, MA, USA).

The absorbance of palladium nanoparticles was measured using a SINCO S-3100 UV-vis absorption spectrophotometer (SINCO, Tokyo, Japan). TEM images of palladium nanoparticles were obtained by JEOL JEM-2010 transmission electron microscopy (300 kv, Tokyo, Japan) after placing a drop of hydrosol on a carbon-coated Cu grid (200 mesh). XRD patterns using Cu Kα (0.1542 nm) radiation were obtained using a Bruker AXS D8 DISCOVER diffractometer (XRD, Bruker, Berlin, Germany). The amount of palladium to be measured in electrochemistry was measured using inductively coupled plasma mass spectrometry (ICP-MS, Agilent 7700S, Santa Clara, CA, USA). 

The electrochemical properties of palladium were measured using a CH Instruments model 708C potentiostat (CHI 780C, Austin, TX, USA), and the counter and reference electrodes were measured with a system of three-electrode cells consisting of Pt wire, Ag/AgCl (saturated 3M NaCl, CHI 111, Austin, TX, USA) and glass carbon electrodes (GCE, CHI131, Austin, TX, USA), respectively. Prior to loading the catalyst on the working electrode of 3 mm GCE, the electrode was polished with alumina powder and thoroughly washed with Milli-Q water and ethanol. 4 μL (metal loading 1 μg: 0.25 mg mL^−1^) of palladium catalyst solution was dropped into the GCE before performing the CV measurement. After drying this sample, a 0.05 wt% Nafion solution (4 μL) was dropped on the surface and dried in an oven at 50 °C, and this GCE was washed several times with acetone, water and ethanol. The electrode was electrochemically cleaned with 50 potential cycles at a scan rate of 50 mV s^−1^ between −0.8 and 0.3 V compared to Ag/AgCl in alkaline electrolyte solution in order to remove the capping agent remaining on the catalyst surface (KOH). The electrochemically active surface area (ECSA) was estimated by the following equation. ECSA = Qo/qo, where Qo is the surface charge achievable in the region under the CV trace of oxygen reduction and qo is the oxygen monolayer reduction in Pd [15,16] (420 μC cm^−2^).

## 3. Results and Discussion

### 3.1. Structure Characterization of Catalyst Materials

As described above, the PdNPs were prepared from aqueous solutions of K_2_PdCl_4_ and CTAC, wherein CTAC acted as both a reductant and a surfactant. During preparation of the NPs, the reducing agent and the capping agent were incorporated into the same molecule of poly(diallyl dimethylammonium) and cetyltrimethylammonium bromide (CTAB) under high temperature conditions [17]. Upon comparison of the FT-IR spectra of the original CTAC and the CTAC-stabilized PdNPs, it could be seen that nitro groups were formed after the oxidation reaction of CTAC (Figure 1). Even though the IR peaks of the stabilized CTAC-PdNPs shifted slightly from the original peak positions for CTAC, the two spectra were similar. However, a new peak was observed at 1388 cm^−1^ in the spectrum of the CTAC-Pd NPs, which was assigned to the N=O vibration. This indicates that the nitroso group formed upon the oxidation of CTAC. The groups to which each peak could be assigned positions are summarized in Appendix A.

Importantly, the size and shape of the PdNPs could be controlled by adjusting the CTAC concentration. More specifically, NPs measuring 17.6 ± 2.3, 40.3 ± 3.2, 43.3 ± 2.8 and 51.4 ± 3.3 nm were observed by TEM, and these were assigned as Pd10, Pd30, Pd50 and Pd80, based on the CTAC concentrations employed during their preparation (Figure 2). Interestingly, the Pd10 NPs were spherical in shape, while the Pd30, Pd50 and Pd80 NPs adopted a multi-pod configuration (see insets of the TEM images in Figure 2). In terms of their size variation, measurement of the branches of the multi-pod Pd NPs w by TEM, gave NP lengths and thicknesses of 12.8 ± 3.2, 17.3 ± 3.0 and 21.2 ± 3.6 nm and 4.1 ± 0.8, 3.8 ± 1.1 and 4.2 ± 1.2 nm, for Pd30, Pd50 and Pd80, respectively.

The optical properties of the various PdNPs were then evaluated by UV-Vis spectrophotometry, and it was found that different properties were observed for the two different shapes of NPs (Figure 3a). However, no significant differences were observed between the various multi-pod NPs since the structure itself, rather than its size, is responsible for the surface plasmon resonance (SPR) in the extinction spectrum. Formation of the PdNPs took place by reaction of the metal ions with the reducing surfactants, although the observation of a peak at 407 nm in the UV-Vis spectrum corresponds to unreduced Pd(II) [18]. In contrast, the observation of a peak at 280 nm indicates the successful reduction of Pd(II).

The ability to control of the sizes and shapes of the PdNPs is known to depend on the seeds employed, the capping agent, the reducing agent and the presence of any additional metal ions in the reaction medium. More specifically, we found that control of the size and shape of the PdNPs was possible by controlling the rate of the reduction reaction, which was in turn controlled by varying the concentration of CTAC. It was confirmed that the NP size increased as the concentration of the surfactant increased (Figure 2). In the XRD patterns of the four prepared PdNP samples, two diffraction peaks were observed in the range of 30° < 2θ < 60°, which were indexed to the (111) and (200) planes of the face centered cubic (fcc) structure of metallic Pd (Figure 3b). According to Bragg’s law, we found that the (111) plane of palladium measures 2.24 Å [19], which is consistent with the measurements made by HR-TEM (Figure 2b,d,f,h).

### 3.2. Electrochemical Characterisation of Catalyst Materials

Pd NPs are known to exhibit an efficient electrocatalytic activity in the ethanol oxidation reaction under alkaline electrolyte conditions [20]. Furthermore, the mechanism of ethanol oxidation of palladium nanoparticles has been well reported. We therefore investigated the electrocatalytic activities of the various prepared PdNPs in this reaction [21]. Thus, Figure 4a shows the CV profiles for this reaction when the various PdNPs were employed as electrodes in a 0.1 M KOH electrolyte solution with scan rate of 50 mV s^−1^. The current densities of the PdNPs were normalized to the ECSA, which was calculated by measuring the coulombic charge for oxygen desorption [8]. Electrochemical measurements showed specific anodic peaks in the forward and reverse sweeps for the various PdNPs during the reaction. In the forward scan, it was observed that the magnitude of the anodic peak current increased significantly for the PdMNPs compared to the PdSNPs. The ECSA-normalized current densities and the corresponding mass activities were 2.80, 6.56, 4.74 and 4.57 mA cm^−2^ and 480, 1423, 1021 and 616 mA mg^−1^ for the responses of the Pd10, Pd30, Pd50 and Pd80 NPs, respectively (Figure 4b,d and Appendix A). These results indicated that compared to the spherical Pd10 NPs, the anodic peak current of the Pd30 NPs was 2.34 times higher in terms of the ECSA-normalized current density and 2.96 times higher in terms of the mass density current. For the ethanol oxidation reaction, this clearly indicates that Pd30 exhibited a significantly improved electrocatalytic activity compared to Pd10, and this was attributed to the multi-pod shape exhibiting a larger surface area and a higher surface energy than the spherical structure. Finally, to test the stabilities of the palladium catalysts, we conducted CA measurements at −0.1 V vs. Ag/AgCl, and our results indicated that the electrochemical stability and distribution per 100s of Pd30 is superior to that of Pd10 in the ethanol oxidation reaction (Figure 4c). Furthermore, Pd30 were stable as result of TEM images CA before and after CA measurement (Appendix A). TEM image of results of the measurement of the Pd30 nanoparticles CA before and it was found that stable.

## 4. Conclusions

Following the successful preparation of a series of Pd nanoparticles (PdNPs), it was found that their shape and size could be controlled by simply varying the concentration of cetyltrimethylammonium chloride at high temperatures. Both the spherical and multi-pod NPs demonstrated electrochemical properties in the ethanol oxidation reaction, although the multi-pod PdNPs exhibited superior activities due to their high surface areas and surface energies. This work is expected to be applicable in the development of new fuel cells for the ethanol oxidation reaction.

## Figures and Tables

**Figure 1 materials-14-02970-f001:**
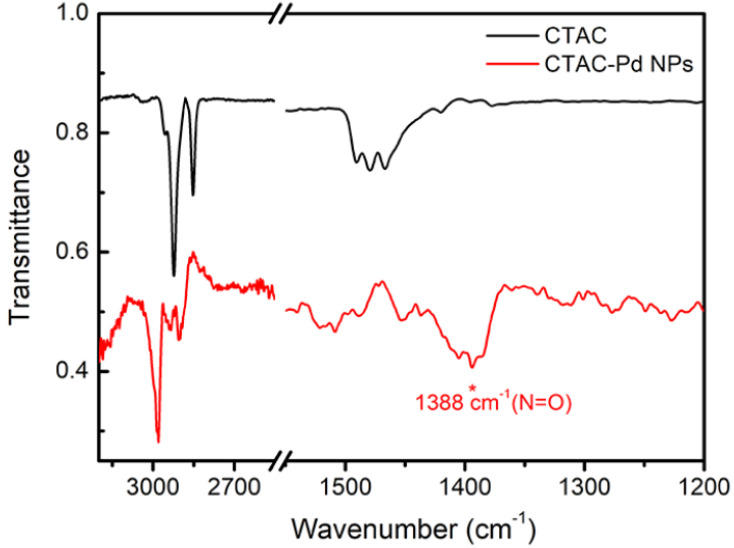
FT-IR spectra of CTAC and the CTAC-Pd NPs.

**Figure 2 materials-14-02970-f002:**
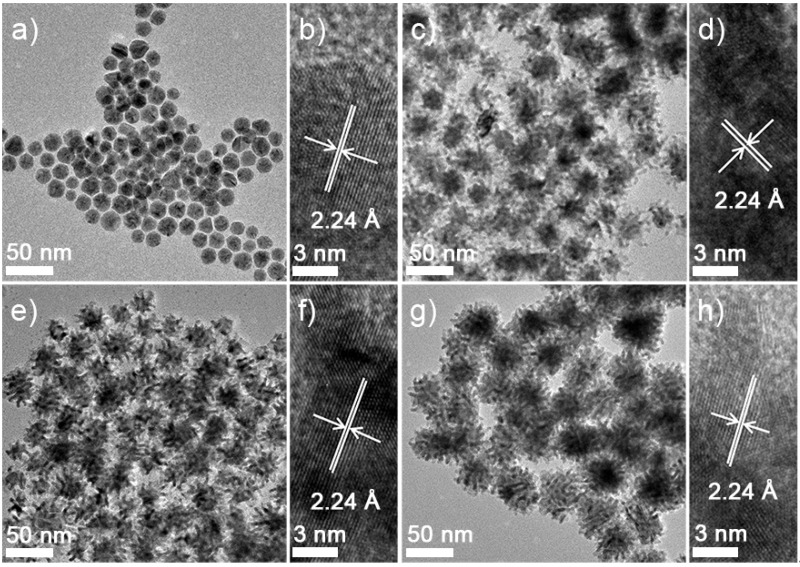
TEM images of the PdNPs in solution: (**a**,**b**) Pd10 with high magnification, (**c**,**d**) Pd30 (**e**,**f**) Pd50 with high magnification, and (**g**,**h**) Pd80 with high magnification.

**Figure 3 materials-14-02970-f003:**
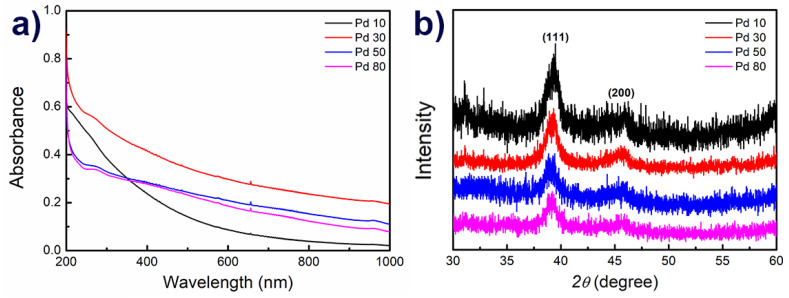
UV-Vis spectra (**a**) and XRD data (**b**) for the Pd NPs of Pd10, Pd30, Pd50, and Pd80.

**Figure 4 materials-14-02970-f004:**
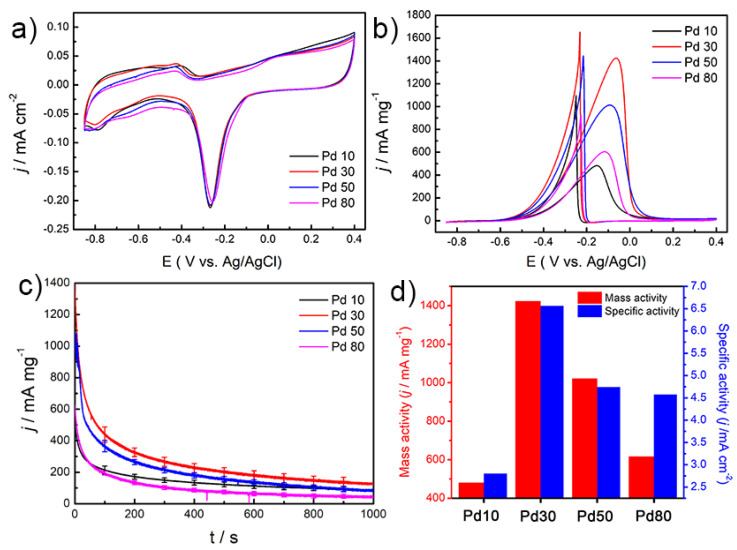
CV of (**a**) Pd NPs on carbon electrodes with a scan rate of 50 mV s^−1^ in 0.1 M KOH. (**b**) CV for the ethanol electrooxidation reaction on different electrodes in 0.5 M ethanol +0.5 M KOH with scan rate of 50 mV s^−1^. (**c**) Chronoamperometric curves and the error bars based standard deviation for the ethanol electrooxidation reaction at −0.1 V versus Ag/AgCl on a Pd NP electrode for Pd10, Pd30, Pd50 and Pd80 in a 0.5 M ethanol +0.1 M KOH solution. (**d**) Current performances for ethanol oxidation on the four other sizes of catalyst.

## Data Availability

The data is available on the request to the corresponding author.

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
