# Peer review of "Shape- and Size-Controlled Palladium Nanocrystals and Their Electrocatalytic Properties in the Oxidation of Ethanol"

_materials, 2021, doi:10.3390/ma14112970_

Round 1

Reviewer 1 Report

It's a good paper for the size- and shape-controlled Pd NPs synthesis. It can be published after the following issues are addressed. 

  1. Line 145. It should be Figure 4.
  2. I don't think it's appropriate to show the electrochemical stability by using a 1000-seconds chronoamperometric measurement. If Pd NPs stability can change with 1000s operation, the applicability of this material is quite questionable. A lot of contingency factors can affect the final equilibrium current, like the falling-off catalysts, heterogeneous distribution of Nafion film. If the authors want to claim that the CA test can be used to show the difference of stability, multiple running results with standard deviation have to be provided. 
  3. A suggestion: Since the author believes that the reduction rate which is controlled by CTAC concentration can affect the morphology of NPs, the semi-batch synthesis method that adds CTAC slowly to the K2PdCl4 solution may be a way to do the synthesis. 

Author Response

Response to Reviewer 1 Comments

Reviewer 1

Point 1: Line 145. It should be Figure 4

Response 1: Thank you for your kind comment. In Line 145, the paper has been revised to Figure 4.

Point 2: I don't think it's appropriate to show the electrochemical stability by using a 1000-seconds chronoamperometric measurement. If Pd NPs stability can change with 1000s operation, the applicability of this material is quite questionable. A lot of contingency factors can affect the final equilibrium current, like the falling-off catalysts, heterogeneous distribution of Nafion film. If the authors want to claim that the CA test can be used to show the difference of stability, multiple running results with standard deviation have to be provided

Response 2: Thank you for your kind comment. In the chronoamperometric curves in Figure 3, the error bars based standard deviation per 100 seconds has been added.

Figure 4. CV of (a) Pd NPs on carbon electrodes with a scan rate of 50 mVs−1 in 0.1 M KOH. (b) CV for the ethanol electrooxidation reaction on different electrodes in 0.5 M ethanol + 0.5 M KOH with scan rate of 50 mVs−1. (c) Chronoamperometric curves and the error bars based standard deviation for the ethanol electrooxidation reaction at −0.1 V versus Ag/AgCl on a Pd NP electrode for Pd10, Pd30, Pd50, and Pd80 in a 0.5 M ethanol + 0.1 M KOH solution. (d) Current performances for ethanol oxidation on the four other sizes of catalyst.

Point 3: A suggestion: Since the author believes that the reduction rate which is controlled by CTAC concentration can affect the morphology of NPs, the semi-batch synthesis method that adds CTAC slowly to the K2PdCl4 solution may be a way to do the synthesis. 

Response 3: Thanks for your suggestion. It was synthesized by slowly adding a Pd solution using the Semi-Batch synthesis method you suggested. When the synthesized palladium nanoparticles were measured by SEM, it was found that CTAC has an effect.

Figure. SEM images of Pd10, Pd30, Pd50 and Pd80

Finally, we deeply appreciate the reviewer once again for giving very critical comments. Thank you.

Reviewer 2 Report

Lee et al. reported Pd nanoparticles and its electrocatalytic properties. The results show that the size and shape are crucial factors to electrocatalytic activities toward the ethanol oxidation reaction. However, Pd nanoparticles as electrocatalyst is lack of novelty and the characterizations are not enough to support the hypothesis. The authors must improve this manuscript and investigate the relevant properties in detail. I do not recommend this manuscript published on materials. The specific comments are shown below:

  1. The novelty of Pd NPs in this study should be described.
  2. In Figure 2, the authors mentioned the size, length, and thickness of nanoparticles. Please provide the related distribution charts. BET analysis should also be provided to prove its high surface areas.
  3. Please provide XPS analysis of the as-synthesized sample and check the chemical state.
  4. Please clearly describe the absorption behavior of Pd NPs and give more details (Figure 3(a)).
  5. The authors should give the mechanism of the electrocatalytic reaction.
  6. Stability study should be provided.
  7. The author should check the figure caption of Figure 2 and Figure 3. Some grammar mistakes and typos should be corrected. In the Title, " Shape- and Size-Controlled Palladium Nanocrystalsand…."

Author Response

Reviewer 2

Point 1: The novelty of Pd NPs in this study should be described. 

Response 1: Thank you for your kind comment. In this study, since the shape and size of palladium nanoparticles can be controlled using only the concentration of surfactant and aqueous solution of K2PdCl4, the shape control and size control are easier and simpler than other synthetic methods. When the synthesized palladium nanoparticles are used an electrochemical catalyst, it has novelty about the characteristics of the size and shape of the palladium nanoparticles.

Point 2: In Figure 2, the authors mentioned the size, length, and thickness of nanoparticles. Please provide the related distribution charts. BET analysis should also be provided to prove its high surface areas.(BET)

Response 2: Thank you for your kind comment. BET is a good measurement method to measure the surface area of a solid catalyst. However, in electrochemical reactions, ECSA is generally measured rather than BET in order to measure the surface area of a metal catalyst. We present the relevant measurement methods to the ECSA and the results regarding them in the text. It is generally accepted that the reduction peaks observed between -0.4 and -0.1 V vs. Ag.AgCl at negative scan are attributed to reduction of oxygen species adsorbed on the surface of catalysts [Adv. Mater. 2017, 1703057]. In addition, the method to calculate ECSA based reduction peak areas were reported in previously references [ref. Woods, R. In Electroanalytical Chemistry: A Series of Advances (vol.9); Bard, A. J., Ed.; Marcel Dekker: New York, 1974; pp 1-162]. To clarify, we changed “CV trace of oxygen reduction” to “CV trace of oxygen species reduction as below. In electrochemistry, the surface area of the catalyst can be determined using ECSA instead of BET. Using the BET ECSA instead in an electrochemical it can be seen that the surface area of the catalyst.

- The (ECSA) was estimated by the following equation: ECSA = Qo/qo, where Qo is the surface charge that can be obtained from the area under the CV trace of oxygen species reduction, and qo is the charge required for the reduction of a monolayer of oxygen on Pd (420 μC cm−2).

Point 3: Please provide XPS analysis of the as-synthesized sample and check the chemical state.

Response 3: Thank you for your kind comment.The influence of the XPS data of palladium nanoparticles is used as the effect on the binding energy to the catalyst by d-orbital movement of palladium in the case of nanoparticles made of alloys of more than bimetallic [1-3]. In this paper, it can be said that the shape and size of palladium have a large effect on the surface area than the XPS data

[1] Wang, Y., Shi, F. F., Yang, Y. Y., & Cai, W. B. (2013). Carbon supported Pd–Ni–P nanoalloy as an efficient catalyst for ethanol electro-oxidation in alkaline media. Journal of power sources, 243, 369-373.

[2] Kadirgan, F., Beyhan, S., & Atilan, T. (2009). Preparation and characterization of nano-sized Pt–Pd/C catalysts and comparison of their electro-activity toward methanol and ethanol oxidation. International journal of hydrogen energy, 34(10), 4312-4320.

[3] Li, L., Chen, M., Huang, G., Yang, N., Zhang, L., Wang, H., ... & Gao, J. (2014). A green method to prepare Pd–Ag nanoparticles supported on reduced graphene oxide and their electrochemical catalysis of methanol and ethanol oxidation. Journal of Power Sources, 263, 13-21.

Point 4: Please clearly describe the absorption behavior of Pd NPs and give more details (Figure 3(a)).

Response 4: Thank you for your kind comment. In Uv-vis spectra of different catalysts, absorption intensity of all catalysts catalyst decreases from short wavelength (200 nm) to long wavelength (1000 nm). However, for case of Pd10, their absorption intensity was more dramatically decreased compared with Pd 30, Pd 50, and Pd 80. This might be attributed to branches of Pd 30, Pd 50, and Pd 80. The branches can induce more complex interaction including surface plasmon resonance between NP and incident light. Therefore, we believe that distinctive optical properties were observed between Pd 10 and other Pd catalysts.

Point 5: The authors should give the mechanism of the electrocatalytic reaction.

Response 5: Thank you for your kind comment. The details of the mechanism of the ethanol oxidation spot are shown below. Reaction pathways for interfacial CH3CH2OH at Pd electrodes in alkaline media. Reproduced with permission from Reference [1].

[1] Yang, Y.-Y.; Ren, J.; Li, Q.-X.; Zhou, Z.-Y.; Sun, S.-G.; Cai, W.-B. Electrocatalysis of ethanol on a Pd electrode in alkaline media: An in situ attenuated total reflection surface-enhanced infrared absorption spectroscopy study. ACS Catal. 2014, 4, 798–803.

The C1 pathway is the complete oxidation of ethanol to CO2 or carbonates via COads intermediate by delivering 12 electrons and the C2 pathway is the partial oxidation of ethanol to acetate by delivering four electrons or to acetaldehyde by delivering two electrons without the breaking of the C–C bond as shown in the following equations [2].

[2] Wang, Y., Zou, S., & Cai, W. B. (2015). Recent advances on electro-oxidation of ethanol on Pt-and Pd-based catalysts: From reaction mechanisms to catalytic materials. Catalysts, 5(3), 1507-1534.

Point 6: Stability study should be provided. 

Response 6: Thank you for your kind comment. It is described as chronoamperometric curves in Figure 4c.

Point 7: The author should check the figure caption of Figure 2 and Figure 3. Some grammar mistakes and typos should be corrected. In the Title, "Shape- and Size-Controlled Palladium Nanocrystalsand…."

Response 7: Thank you for your kind comment. The paper has been revised to caption of Figure 2 and Figure 3. 

Finally, we deeply appreciate the reviewer once again for giving very critical comments. Thank you.

Reviewer 3 Report

The reviewed manuscript is dedicated to shape- and size-controlled synthesis of Pd NPs and their application as electrocatalysts for ethanol electrooxidation. In general, the manuscript is logically written and interesting from scientific point of view.

I recommend the manuscript to be published in Materials after minor revisions:

  1. The statement about the importance of size and shape of Pd NPs is mentioned a few times in the introduction. Such repetition in short article is not necessary.
  2. Why Nafion was added separately and not directly as constituent part of slurry (ink)?
  3. The Figure 1a shows more differences as mentioned in the main text. Please extend the comparison of FT-IR spectra in more details.
  4. What is “SPR”? (line 113)
  5. The Figure 4c clearly reveals pronounced drop in current density within 30-40 s (six times lower j). What is the origin of this?
  6. Time of 1000s is very short stability experiment. Do you have really long-term chronoamperometry?
  7. Did you characterize the material after carried out EC experiments?
  8. Some comments and remarks concerning formatting of written manuscript:
  • there is no term “crystal phase structure”;
  • the resistivity value for milipore water is missed (line 53);
  • it is not correct to write AgCl RE, it is Ag/AgCl;
  • the information about the electrolyte purging (line 70) is interrupting the description of WE preparation;
  • link to Figure 1 in line 103 looks not proper one;
  • Line 105: Figure 2 (instead of Figure 1);
  • Line 145: in manuscript there is no mentioned Figure 5;
  • Line 156: vs. Ag/AgCl instead of Ag/AgNO3;
  • Line 157: formic acid electrooxidation is not the topic of this work;
  • Line 160: at -0.1 V (instead of 0.1 V).

Author Response

Reviewer 3

Point 1: The statement about the importance of size and shape of Pd NPs is mentioned a few times in the introduction. Such repetition in short article is not necessary.

Response 1: Thank you for your kind comment. Lines 28-30 have been removed from this paper. (In this context, direct alcohol and formic acid 28 fuel cells have been applied in diverse electronic portable devices, such as laptops, cell-29 phones, and electric vehicles [5–7].)

Point 2: Why Nafion was added separately and not directly as constituent part of slurry (ink)?

Response 2: Thank you for your kind comment.The reason for adding nafion during slurry preparation is that in general, the direct ethanol fuel cell (DEFC) has a low operating temperature, high power density, a clear phase separation between the hydrophilic and hydrophobic regions, and the polymer is flexible to form an ion aggregate well. Nafion, which has excellent mechanical strength, is most often used.

Point 3: The Figure 1a shows more differences as mentioned in the main text. Please extend the comparison of FT-IR spectra in more details. 

Response 3: Thank you for your kind comment. In order to show IR peak in detail, only the IR peak was modified, and the CTAC table was modified to Table 1 of the supplementary data.

Point 4: What is “SPR”? (line 113)

Response 4: Thank you for your kind comment. SPR stands for Surface Plasmon Resonance, and was added to this paper

Point 5: The Figure 4c clearly reveals pronounced drop in current density within 30-40 s (six times lower j). What is the origin of this?

Response 5: Thank you for your kind comment. The reason for the rapid decrease in the current density in Fig. 4c is that the surface area of palladium nanoparticles decreases as CO is adsorbed by the generation of CO during the formation of intermediates in the forword of the ethanol oxidation reaction.

Point 6: Time of 1000s is very short stability experiment. Do you have really long-term chronoamperometry?

Response 6: Thanks for your comment. Chronoamperometric curves are widely used as catalyst stability experiments, and 1000 seconds is a short time, but it is used in several papers, so a reference is attached[1-2].

[1] Alfi, N., Rezvani, A. R., Khorasani-Motlagh, M., & Noroozifar, M. (2017). Synthesis of europium oxide-promoted Pd catalyst by an improved impregnation method as a high performance catalyst for the ethanol oxidation reaction. New Journal of Chemistry, 41(19), 10652-10658.

[2] Wang, X. M., & Xia, Y. Y. (2009). Synthesis, characterization and catalytic activity of an ultrafine Pd/C catalyst for formic acid electrooxidation. Electrochimica acta, 54(28), 7525-7530.

Point 7: Did you characterize the material after carried out EC experiments?

Response 7: Thanks for your comment. TEM images before and after chronoamperometric curves were added to the supplementary data.

Figure S2. TEM images of Pd 30 catalysts (a) before and (b) after Chronoamperometric curves. 

Point 8: Some comments and remarks concerning formatting of written manuscript: 

Response 8: Thanks for your comment. The following content has been revised in red text in this paper.

  • there is no term “crystal phase structure”;
  • the resistivity value for milipore water is missed (line 53);
  • it is not correct to write AgCl RE, it is Ag/AgCl;
  • the information about the electrolyte purging (line 70) is interrupting the description of WE preparation;
  • link to Figure 1 in line 103 looks not proper one;
  • Line 105: Figure 2 (instead of Figure 1);
  • Line 145: in manuscript there is no mentioned Figure 5;
  • Line 156: vs. Ag/AgCl instead of Ag/AgNO3;
  • Line 157: formic acid electrooxidation is not the topic of this work;
  • Line 160: at -0.1 V (instead of 0.1 V).

Finally, we deeply appreciate the reviewer once again for giving very critical comments. Thank you.

Reviewer 4 Report

Thanks for the invitation to review the manuscript where authors synthesized Pd NCs and used it to oxidise ethanol. The manuscript requires major revison before publication

  1. Introduction is not clear neither it has been discussed why noble metals are necessary.
  2. In CV authors have irreverisble redox peak which might be due to the oxygen reduction reaction as well. Did authors clarify it.
  3. Calculation regarding ECSA should be provided. What is the stability of catalyst.
  4. Important references should be included ChemCatChem, 2019, 11, 4383– 4392; https://doi.org/10.1016/j.apsusc.2020.146266.

Author Response

Point 1: Introduction is not clear neither it has been discussed why noble metals are necessary.

Response 1: Thank you for your constructive comment. To improve the importance of PdNPs, we added related sentences as below.

-“In particular, PdNPs are widely used as an electrocatalyst for anode reactions of fuel cell systems, which are clean and sustainable energy converting system, generating electricity form chemical fuels. However, several factors such as high cost and poor stability hinder the practical application of fuel cell systems. Recently, PdNPs with controlled shape and size showed enhanced electrocatalytic performances for electrochemical fuel oxidations, which signifies the importance of preparation of PdNPs with desirable shapes and sizes.”

Point 2: In CV authors have irreverisble redox peak which might be due to the oxygen reduction reaction as well. Did authors clarify it.

Response 2: Thank you for your kind comment. It is generally accepted that the reduction peaks observed between -0.4 and -0.1 V vs. Ag.AgCl at negative scan are attributed to reduction of oxygen species adsorbed on the surface of catalysts [Adv. Mater. 2017, 1703057]. In addition, the method to calculate ECSA based reduction peak areas were reported in previously references [ref. Woods, R. In Electroanalytical Chemistry: A Series of Advances (vol.9); Bard, A. J., Ed.; Marcel Dekker: New York, 1974; pp 1-162]. To clarify, we changed “CV trace of oxygen reduction” to “CV trace of oxygen species reduction as below.

- The electrochemically active surface area (ECSA) was estimated by the following equation: ECSA = Qo/qo, where Qo is the surface charge that can be obtained from the area under the CV trace of oxygen species reduction, and qo is the charge required for the reduction of a monolayer of oxygen on Pd (420 μC cm−2).

Point 3: Calculation regarding ECSA should be provided. What is the stability of catalyst.

Response 3: Thank you for your kind comment. As suggested by the reviewer, we added a following sentence in the references and notes section for explaining the ECAS method used for the normalization of current densities.

“(2) The ECAS was estimated by the following equation; ECAS = Qo/qo, where Qo is the surface charge that can be obtained from the area under the CV trace of oxygen desorption and qo is the charge required for desorption of monolayer of oxygen on the Pd surface [1] (420 μC/cm2).”

(1)         Woods, R. In Electroanalytical Chemistry: A Series of Advances (vol.9); Bard, A. J., Ed.; Marcel Dekker: New York, 1974; pp 1-162.

(2)         The ECAS was estimated by the following equation; ECAS = Qo/qo, where Qo is the surface charge that can be obtained from the area under the CV trace of oxygen desorption and qo is the charge required for desorption of monolayer of oxygen on the Pd surface1 (420 μC/cm2).

Point 4: Important references should be included ChemCatChem, 2019, 11, 4383– 4392; https://doi.org/10.1016/j.apsusc.2020.146266. 

Response 4: Thank you for your kind comment. Pd NPs are known to exhibit an efficient electrocatalytic activity in the ethanol oxidation reaction under alkaline electrolyte conditions. We therefore investigated the electrocatalytic activities of the various prepared PdNPs in this reaction [18].

(18)       Singh, P., Gangadharan, P. K., Khan, Z., Kurungot, S., & Jaiswal, A. (2019). Cubic palladium nanorattles with solid octahedron gold core for catalysis and alkaline membrane fuel cell Applications. ChemCatChem, 11, 4383-4392.

Finally, we deeply appreciate the reviewer once again for giving very critical comments. Thank you.

Round 2

Reviewer 4 Report

Authors have revised well the manuscript. Therefore, it can be accepted for publication.